# Reduced HLA-I Transcript Levels and Increased Abundance of a CD56^dim^ NK Cell Signature Are Associated with Improved Survival in Lower-Grade Gliomas

**DOI:** 10.3390/cancers17091570

**Published:** 2025-05-05

**Authors:** Md Abdullah Al Kamran Khan, Lorenza Peel, Alexander J. Sedgwick, Yuhan Sun, Julian P. Vivian, Alexandra J. Corbett, Riccardo Dolcetti, Theo Mantamadiotis, Alexander D. Barrow

**Affiliations:** 1Department of Microbiology and Immunology, The University of Melbourne at The Peter Doherty Institute for Infection and Immunity, Melbourne, VIC 3000, Australia; 2St. Vincent’s Institute of Medical Research, Melbourne, VIC 3065, Australia; 3Department of Medicine, The University of Melbourne, Melbourne, VIC 3000, Australia; 4Australian Catholic University, Melbourne, VIC 3065, Australia; 5Peter MacCallum Cancer Centre, Melbourne, VIC 3000, Australia; 6Sir Peter MacCallum Department of Oncology, The University of Melbourne, Melbourne, VIC 3000, Australia; 7Department of Surgery, Royal Melbourne Hospital, The University of Melbourne, Melbourne, VIC 3000, Australia

**Keywords:** lower-grade glioma, prognosis, NK cells, CD56^dim^ NK, HLA-I, anti-tumour immunity, transcriptional signature, deconvolution

## Abstract

Human leukocyte antigen class I (HLA-I) molecules can modulate anti-tumour immune responses from CD8^+^ T cells and NK cells. However, how deregulated HLA-I expression impacts clinical outcomes in cancer patients has remained unclear. Using computational approaches, we investigated the association of HLA-I molecules with patient survival by analysing gene expression datasets across multiple cancers. We observed a trend toward poor survival in patients with high HLA-I expression in lower-grade gliomas. Moreover, the favourable prognostic association of CD56^dim^ NK cells was attenuated in the context of abundant HLA-I, as suggested by the correlation between NK cell receptors NKG2A/C/E and HLA-E. Overall, our study provides a computational framework that offers insights into HLA-I-mediated modulation of cytotoxic NK cell activity using cancer gene expression datasets, with potential applicability for other diseases.

## 1. Introduction

Lower-grade gliomas (LGGs) are neoplastic transformations of the supporting glial cells (i.e., astrocytoma, oligodendroglioma, and oligoastrocytoma) of the central nervous system (CNS) [1]. They account for 15–20% of all primary brain cancers, with an estimated global incidence of 0.25 to 0.75 cases per 100,000 individuals annually [2,3]. Immune cells infiltrate the glioma microenvironment, influencing tumour development and patient prognosis [4,5]. Previous studies have highlighted a reduced tumour infiltration of T cell subsets in LGG compared to high-grade gliomas (HGGs) [6,7,8]. A similar pattern has also been observed for different tumour-infiltrating myeloid cells such as microglia and monocyte-derived macrophages [6,9,10]. Though the presence of different T cell subsets and tumour-associated macrophages (TAMs) in various glioma types has been extensively studied, the role of NK cell subsets and MHC class I in LGGs remains under-reported.

Human leukocyte antigen class I (HLA-I) molecules are fundamental regulators of CD8^+^ T cell- and NK cell-mediated immune surveillance in the tumour microenvironment (TME) [11,12]. The classical HLA-I molecules, namely HLA-A, -B, and -C, are highly polymorphic, whilst the non-classical HLA-I molecules HLA-E, -F, and -G are less so [13]. The classical HLA-I molecules, complexed with β2-microglobulin (β2m), present peptide antigens to CD8^+^ T cells [11]. In the TME, tumour cells use a range of mechanisms to downregulate classical HLA-I molecules, primarily to evade recognition and killing by CD8^+^ T cells. However, this makes tumours vulnerable to NK cell-mediated killing [14]. NK cells monitor cells for the loss of classical HLA-I expression through inhibitory receptors that can either prime NK functional potential, e.g., NK “education” or NK “licensing”, during NK cell development, or restrain the functional responses of mature ‘educated’ NK cells (“missing self”) [15,16,17,18,19]. The non-classical HLA-I molecules, HLA-G and HLA-E, play critical roles in modulating NK cell responses via LILR receptors (through binding to HLA-G and HLA-E) and the CD94/NKG2x (i.e., NKG2A, NKG2C, and NKG2E) receptors (through binding to HLA-E) [14,20,21,22,23]. Thus, tumour cells often downregulate classical HLA-I molecules whilst upregulating the expression of non-classical HLA-I molecules, especially HLA-E, to evade both T cell and NK cell surveillance [22,24].

Traditionally, NK cells are classified into two functional subtypes, the potently cytotoxic but weakly cytokine-producing CD56^dim^ NK cell subset and the strongly cytokine-producing, less cytotoxic CD56^bright^ NK cell subset [25]. NK cells exert anti-tumour effects through direct cytolytic killing of tumour cells, as well as by mediating immune responses via the release of pro-inflammatory cytokines, such as IFN-γ and TNF, which promote tumour cell apoptosis, suppress angiogenesis, and modulate the recruitment and function of other immune cells in the TME [26,27,28]. The infiltration and functional relevance of NK cells in solid tumours, including gliomas, have been reported previously [28,29,30]. Importantly, the potential roles of NK cells in brain cancers, particularly in gliomas, have also been discussed [31,32,33,34,35]. NK cells are thought to contribute to eliminating early-stage cancers as well as glioblastoma (GBM) stem cells [36,37]. NK cells may also exert neuroprotective effects by modulating microglial function, thereby suppressing other inflammatory cells and clearing toxic aggregates [36,38]. Glioma-infiltrating NK cells exhibit reduced levels of NKG2D, while glioma tissues downregulate the expression of NKG2D ligands [36,39]. Additionally, GBM cells can curb NK cell-mediated immunosurveillance through elevated expression of inhibitory molecules such as the class-I HLAs, lectin-like transcript 1 (LLT1), regeneration and tolerance factor (RTF), and growth/differentiation factor-15 (GDF15) [36]. Despite these insights, the impact of HLA-I expression on tumour-infiltrating NK cells in LGGs remains largely unexplored. In this study, we employed a computational approach to investigate the impact of altered tumour HLA-I expression on the transcriptional signatures of CD56^dim^ and CD56^bright^ NK cells in LGGs. Our results reveal an inverse relationship between the expression of HLA-I and the transcriptional signature of CD56^dim^ NK cells in LGGs, where LGG patients with low HLA-I expression and high levels of CD56^dim^ NK cells have better survival probabilities.

## 2. Materials and Methods

### 2.1. Retrieval of LGG Patient Transcriptomes from TCGA and Chinese Glioma Genome Atlas (CGGA) Databases

We retrieved the patient RNA-seq datasets of 28 solid TCGA tumours and tumour-adjacent normal tissues along with the patients’ clinical information from the GDC data portal [40], and the CGGA LGG patient RNA-seq datasets (mRNAseq_693 and mRNAseq_325) were obtained from the CGGA data portal (http://www.cgga.org.cn/; accessed on 23 September 2024) [41]. The TCGA LGG dataset includes patients diagnosed with lower-grade gliomas, encompassing WHO grade II and III astrocytomas and oligodendrogliomas [42], as per classifications prior to the 2021 WHO update [43]. Accordingly, we selected CGGA patients with primary tumours classified as WHO grade II or III gliomas to ensure consistency across cohorts. An RNA-seq dataset of healthy brain cortex was obtained from the GTEx database [44] for use as the normal control tissue. After their procurement, the RNA-seq datasets were cleaned by removing any duplicated entries, followed by the TMM (Trimmed mean of M) scale normalisation [45] of the transcript read counts to reduce any unwanted variabilities in the data.

### 2.2. Construction of Transcriptional Signatures for CD56^bright^ and CD56^dim^ NK Cells

To simultaneously construct transcriptional signatures for different immune and stromal cells including the CD56^bright^ and CD56^dim^ NK cells, we first obtained the bulk RNA-seq datasets from 20 immune and 3 stromal cell subsets from the curated human bulk transcriptional catalogue (HBCC) [46]. We then adjusted transcript abundances using cellsig, a multilevel Bayesian noise modelling approach, as outlined previously [46]. Next, these adjusted transcriptomes were utilised in the CIBERSORTx algorithm [47] to generate our target transcriptional signature matrix.

### 2.3. Deconvolution of Bulk RNAseq Datasets to Obtain the Relative Abundance of CD56^bright^ and CD56^dim^ NK Cell Subsets

Utilising the generated transcriptional signature matrix as the reference, we used the CIBERSORT cellular deconvolution program [48] via the tidybulk R package [49] to estimate the relative abundance of the CD56^bright^ and CD56^dim^ NK cell subsets along with other immune and stromal cell types in the LGG tumour bulk transcriptomes. Default parameters were applied.

### 2.4. Estimation of HLA-I Gene Abundance

To estimate the abundance of transcripts encoding HLA-I molecules, we first constructed a gene set for the transcripts encoding HLA-A, -B, -C, -E, -F, -G, and β2m. Next, HLA-I gene set scores were calculated for each LGG RNA-seq sample using the singscore [50] R package.

### 2.5. LGG Prognostic Association

To evaluate the clinical significance of the different cell type-specific signatures and transcript expression, we employed Kaplan–Meier (KM) survival analysis, enumerating the progression-free survival of patients using the survminer R package [51]. For the KM estimates, patients were separated into two groups, i.e., high and low, based on a median split of the analysed variable. The significance of the comparison between KM estimates was calculated using the Mantel–Cox log-rank test [52]. Each Kaplan–Meier survival curve includes the global *p*-value from the log-rank test used to compare the groups. Global *p*-values of the composite KM curves were adjusted using the Benjamini–Hochberg (BH) method (Appendix A).

### 2.6. Single-Cell RNA-seq Data Analysis

Single-cell RNA-seq data of two LGG tissues (tumour grade II) were retrieved from the gene expression omnibus (GEO) GSE182109 dataset [53]. Each dataset was pre-processed and normalised with SCtransform in Seurat [54] following the parameters outlined in a previous study [55]. These data were then integrated using the Harmony algorithm [56]. Upon integration, we searched for neighbouring cells using the shared nearest-neighbour (SNN) graph approach. Clustering was then performed using the Leiden algorithm in igraph with a resolution of 0.1 and 0.3 (for the re-clustering of CD45^+^ cells). SingleR [57] was used for the automated annotation of the CD45^+^ cell clusters with the EncodeBlueprint database as a cell type reference. Also, manual curation for the cluster-specific markers was performed for the immune cell clusters by searching for the significantly (average log2FoldChange > 1 and adjusted *p*-value < 0.05) differentially expressed genes (Appendix A).

### 2.7. Statistical Analysis

Statistical significance for the comparison between the means of two independent groups was assessed using the non-parametric Wilcoxon signed-rank test [58] implemented in R version 4.4.2.

## 3. Results

### 3.1. A Pan-Cancer Screen Reveals an Association Between the Differential Expression of HLA-I Transcripts and Cancer Patient Prognoses

To assess the altered expression of HLA-I molecules in different cancers, we first analysed the differential expression of the transcripts encoding HLA-I molecules in 28 solid cancers from TCGA. In cancers such as lower-grade gliomas (LGGs), glioblastoma (GBM), bladder urothelial carcinoma (BLCA), and kidney renal clear cell carcinoma (KIRC), HLA-I transcripts were significantly upregulated, whereas in uterine carcinosarcoma (UCS) and lung squamous cell carcinoma (LUSC), HLA-I expression was downregulated in the tumour tissues (Figure 1A). Cancers such as breast carcinoma (BRCA), adrenocortical carcinoma (ACC), and lung adenocarcinoma (LUAD) showed variable changes in HLA-I transcript expression (Figure 1A).

Next, we sought to identify tumour types where the expression of HLA-I potentially influences the patient’s survival outcomes. In ACC, BLCA, and KIRC, increased expression of HLA-E was associated with favourable patient survival (Figure 1B). Strikingly, we observed a distinct prognostic association in LGG patients, which was different from all the other cancer types analysed (Figure 1B). In LGGs, increased expression of all HLA-I transcripts was associated with poor patient prognoses (Figure 1B).

We then examined the putative association between HLA-I transcript expression and the abundances of tumour-infiltrating CD8^+^ T cell subsets, as well as CD56^dim^ and CD56^bright^ NK cell subsets. We observed a positive association between HLA-I transcript levels and CD8^+^ effector memory T (T_EM_), as well as with CD8^+^ central memory T (T_CM_) cell abundances in several cancers (Figure 1C). Our analysis also revealed a strong overall negative association between the naïve CD8^+^ T cell signature and HLA-I transcript expression in several cancer types, including LGGs (Figure 1C). Intriguingly, in the LGG patient transcriptomes, we observed little to no association between HLA-I transcript expression and the abundance of CD8^+^ T_EM_ and T_CM_ subsets (Figure 1C). For the NK cell compartment, most cancers studied showed a positive correlation between HLA-I transcript expression and CD56^bright^ NK cell abundance. This was not generally observed for the CD56^dim^ NK cell subset. In fact, in head and neck squamous cell carcinoma (HNSC), LGGs, KIRC, and skin cutaneous melanoma (SKCM), an inverse association between the HLA-I transcript levels and the abundance of the CD56^dim^ NK cell subset was observed (Figure 1C). Notably, HNSC and LGGs were the only cancers where a positive correlation with CD56^bright^ NK and a negative correlation with CD56^dim^ NK cells were both observed. Furthermore, The LGG was unique in showing these association in the absence of a detectable correlation with the CD8^+^ T cell compartment. These findings suggest that LGGs may possess a unique tumour-infiltrating lymphocyte (TIL) signature in which increased expression of HLA-I is associated with a reduced abundance of tumour-infiltrating CD56^dim^ NK cells and a poor prognosis.

### 3.2. Association of HLA-I Transcript Expression with Poor LGG Patient Prognosis Is Dependent on Tumour Grade

We next asked whether this association between HLA-I transcript expression and LGG patient survival varied between the different clinical and histological grades of LGGs. To assess this, we first established the abundance of HLA-I transcripts and observed that HLA-A-, -B-, -C-, -E-, and β2m-encoding genes had comparatively higher expression than HLA-F and -G (Figure 2A,B). Additionally, HLA-I transcript abundance was greater in more advanced grades of gliomas (Figure 2C). Astrocytoma consistently expressed the highest levels of all HLA-I transcripts, followed by oligoastrocytomas, with oligodendrogliomas having the lowest (Figure 2D). We then performed survival analysis, which corroborated the negative association of all HLA-I transcripts with LGG patient outcomes (Figure 2E).

We then stratified the data to determine whether the effect of HLA-I expression on LGG patient survival was evident across different glioma grades. First, we established that LGG patients with tumours of varying clinical–pathological grades had distinct survival outcomes; grade 3 gliomas had lower survival probability, and astrocytomas were observed to be the most aggressive histological glioma subtype (Appendix A). These findings correlated with the expression patterns of HLA-I transcripts (Figure 2F). Patients with oligodendroglioma and oligoastrocytoma with low levels of HLA-I expression showed better survival potential compared to patients with astrocytomas or high levels of HLA-I (Figure 2G). Overall, these findings reveal that the prognosis of different glioma types and different grades tightly correlate with the expression levels of HLA-I transcripts.

### 3.3. Expression of the Transcriptional Signature of CD56^dim^ NK Cells Inversely Correlates with HLA-I Transcript Expression in LGGs

We hypothesised that the overexpression of HLA-I molecules in LGGs is a tumour immune-evasion mechanism to evade cytotoxic NK cells. To assess this, we first estimated the relative abundance of the CD56^bright^ and CD56^dim^ NK cell subset signatures in the LGG transcriptomes together with the abundances of other major immune cells (Figure 3A). We observed no difference between the overall abundance of these two subsets (Figure 3B); however, their abundance inversely correlated with each other (Figure 3C). While the abundance of CD56^bright^ NK cells was higher in grade 3 astrocytomas, CD56^dim^ NK cells were more prominent in both grade 2 and grade 3 oligodendrogliomas (Appendix A).

We explored the prognostic implications of the NK cell subset signatures in LGGs and observed an unfavourable survival association with CD56^bright^ NK cells, as well as a trend towards longer patient survival with higher CD56^dim^ NK cell abundance (Figure 3D). Patients with different tumour grades did not have a distinctive survival association with either NK cell subset. However, favourable survival trends were observed for the CD56^dim^ NK subset in oligodendroglioma and oligoastrocytoma patients, while CD56^bright^ NK cells were associated with a poor oligoastrocytoma prognosis (Appendix A).

In the multivariate analysis, a higher tumour mutation burden (TMB) was observed to be strongly associated with poor survival in LGG patients, while CD56^dim^ NK cell abundance was associated with a trend toward improved outcomes. This trend appeared to be independent of TMB and CD8⁺ T cell subset abundances (Appendix A).

Next, we investigated the association between HLA-I molecules and the NK subset abundances in the tumour tissue. Interestingly, we found that CD56^dim^ NK cells were inversely associated with all HLA-I molecule transcripts, whereas the CD56^bright^ NK cells positively correlated with HLA-I transcripts (Figure 4A). The association between CD56^bright^ NK cells and HLA-I was consistent across both oligodendrogliomas and oligoastrocytomas, but the correlation between HLA-I and CD56^dim^ NK cell abundance was observed only in oligodendrogliomas (Appendix A).

To assess the prognostic implications of these associations in glioma patients, we performed a combinatorial survival analysis. We observed that patients with low HLA-I (i.e., β2m, HLA-A, -B, and -E) expression and high levels of the CD56^dim^ NK signature in the tumour tissue had better survival probability compared to the patients with a low CD56^dim^ NK cell signature and high HLA-I expression (Figure 4B).

As HLA-I transcript expression levels were strongly correlated with each other (Appendix A), we constructed a gene set to estimate their aggregated abundance in patient samples. Assessing the association between the aggregated HLA-I expression and the CD56^bright^ and CD56^dim^ NK signatures, we observed a positive correlation with CD56^bright^ NK cell abundance, while CD56^dim^ NK cell abundance was negatively associated (Figure 4C). This was most pronounced in oligodendroglioma patients (Appendix A). LGG patients with increased overall HLA-I expression had poor survival outcomes (Appendix A), and LGG patients with high overall HLA-I expression and low CD56^dim^ NK abundance had poor survival outcomes compared to patients with low overall HLA-I and increased CD56^dim^ NK cell abundance (Figure 4D). Notably, this was most pronounced in the oligoastrocytoma patient cohort (Appendix A).

CD4^+^ T cells can regulate the activation of NK cells [59]; more importantly, the cooperation between NK cells and CD4^+^ T cells could compensate critically where CD8^+^ T cells are dispensable [60,61,62,63,64]. We assessed the prognostic significance of different T cell subsets and observed helper T cells to be associated with favourable LGG prognoses (Appendix A). More evidently, the favourable prognostic associations of helper T and CD56^dim^ NK cells were observed in a multivariate analysis including HLA-I abundance. This suggests that the potential partnership between CD56^dim^ NK cells and CD4^+^ T cells could be critical especially when the LGG tumours are evading the CD8^+^ T cells.

Overall, these findings suggest that the expression of HLA-I inversely correlates with CD56^dim^ NK abundance in LGGs, and CD56^dim^ NK cell abundance is associated with improved survival probability. These observations support the hypothesis that HLA-I may inhibit the infiltration or functionality of cytotoxic CD56^dim^ NK cells within the glioma tumour microenvironment.

### 3.4. Expression Levels of HLA-I and NK Cell Receptor Transcripts Are Associated with LGG Patient Prognoses

Since a potential association of HLA-I expression with the abundance of NK subsets was evident, we next asked whether the expression of activating NK cell receptors and HLA-I was associated with LGG patient survival. We observed that increased abundance of *KLRC2* (NKG2C), *KLRC3* (NKG2E), *KLRC4* (NKG2F), *KLRF1* (NKp80), *KLRK1* (NKG2D), B3GAT1 (CD57), and *SELL* (CD62L) was associated with favourable LGG patient survival (Figure 5A and Appendix A). Interestingly, patients with a high abundance of these receptor-coding transcripts and high CD56^dim^ NK abundance were predicted to have improved survival potential (Figure 5A and Appendix A), emphasising the possible role of these activating receptors in mediating the anti-tumour functions of CD56^dim^ NK cells in this setting [65]. Furthermore, a negative association between these receptors and HLA-I expression was evident with respect to patient survival, as only the group of patients with high transcript expression of these activating receptors and low HLA-I expression had favourable survival outcomes (Figure 5A and Appendix A).

NKG2x receptors can shape the functional status of NK cells by signalling through the NKG2A (inhibitory) or NKG2C/NKG2E (activating) receptors when they engage HLA-E presented by tumour cells [66]. To assess whether these signalling molecules are present in LGG tissues, we evaluated the associations between HLA-E and NKG2A/NKG2C/NKG2E transcript expression. HLA-E showed a negative correlation with the NKG2C- and NKG2E-coding transcripts (*KLRC2* and *KLRC3*, respectively) (Figure 5B). It is therefore possible that inhibitory HLA-E/NKG2A signalling predominates over the activating NKG2C/E axis in the context of LGG patient survival. Survival analysis also suggested that the patients with low HLA-E but high NKG2C/NKG2E expression may have improved survival potential (Figure 5C). Furthermore, LGG patients harbouring high levels of activating NKG2C/NKG2E/NKG2F receptor transcripts and low inhibitory NKG2A abundance were projected to have better survival outcomes compared to patients with low activating NKG2x and high inhibitory NKG2A transcripts (Figure 5D).

We also evaluated the presence of CD56^dim^ NK cells and the NKG2x receptor-coding genes in single-cell RNAseq data from two LGG patients [53]. Though most of the glioma and brain stroma cells expressed HLA-I genes, immune cells were also enriched for HLA-I genes (Appendix A). The CD56^dim^ NK cell signature was observed in the CD8^+^ T-NK cell cluster (i.e., cluster 5) of the immune cells (Appendix A). Furthermore, cluster 5 exhibited strong enrichment of the CD56^dim^ NK signature compared to other immune cell types (Appendix A). The NKG2x receptor-coding genes were expressed by a fraction of the CD8^+^ T-NK cell cluster cells (Appendix A). Overall, these findings suggest that various NK receptors may have critical roles in dictating the anti-tumour functionalities of the CD56^dim^ NK subset in LGG tissues.

### 3.5. Expression of HLA-I Transcripts Is Negatively Associated with the CD56^dim^ NK Signature in the CGGA LGG Tissue Transcriptomes

Our findings from the TCGA-LGG tissue transcriptome revealed that HLA-I expression may have potential implications for the critical roles of NK cell subsets in clinical outcomes of LGG patients. To further assess and validate these findings, we next investigated the effects of HLA-I expression, NK cell subsets, and NK cell receptors on LGG patient survival using transcriptomic data from the CGGA database. Unlike the TCGA cohort, CGGA LGG patients had a higher abundance of the signature for CD56^bright^ NK cells compared to CD56^dim^ NK cells, but the subset signatures were inversely correlated as observed for the TCGA dataset (Appendix A). Consistent with our previous observations, we detected a favourable survival trend in LGG patients with high expression of the CD56^dim^ NK cell signature (Figure 6A). We did not observe a strong inverse correlation between the aggregated HLA-I expression and the CD56^dim^ NK subset signature as observed for the TCGA cohort, although the negative association was apparent (Figure 6B). Our analysis of the CGGA LGG dataset also revealed the negative impact of increased HLA-I transcript expression on patient survival, as well as the converse effect in patients with high CD56^dim^ NK but low HLA-I abundance (Figure 6C). A high abundance of the signature for helper T cells was associated with favourable CGGA LGG patient survival, and a similar association was also observed for patients with both high levels of helper T and CD56^dim^ NK cells (Appendix A). These favourable associations were also evident in a multivariate analysis with HLA-I abundance in CGGA patients (Appendix A).

We observed that the expression of HLA-I was negatively associated with the expression of NKG2C (*KLRC2*), NKG2E, CD62L (*SELL*), and CD57 (*B3GAT1*), as observed in the TCGA dataset (Figure 6D and Appendix A). Interestingly, the CGGA LGG patients with high NKG2C, NKG2E, CD62L, and CD57 transcript levels along with increased abundance of the CD56^dim^ NK signature were also predicted to have better survival; In addition, patients harbouring high expression of these NK cell receptor-coding genes but low HLA-I expression showed similar survival outcomes (Figure 6D and Appendix A). The expression of HLA-E was positively associated with NKG2A (*KLRC1*) but negatively associated with NKG2C (*KLRC2*) and NKG2E (*KLRC3*) (Appendix A). Survival analysis also showed that patients with abundant NKG2C and NKG2E but low HLA-E had improved LGG survival probability (Appendix A). Additionally, CGGA patients with low abundance of NKG2A but high NKG2C/NKG2E/NKG2F had a better prognosis than those with high NKG2A and low NKG2C/NKG2E/NKG2F levels (Appendix A). Overall, findings from the CGGA LGG patient cohort corroborated our results from the TCGA LGG dataset, supporting the negative impact of HLA-I overexpression on the prognostic implications of glioma-infiltrating NK cells.

## 4. Discussion

Altered expression of HLA-I is frequently observed in solid tumours. However, the underlying effects of altered HLA-I expression on the abundance and functionality of tumour-infiltrating NK cells remain unclear. Previous studies have highlighted the significance of NK cell functional activity against glioblastoma cells in vitro [37,67,68,69,70]. Furthermore, CD56^dim^ CD16^+^ NK cells have been reported to have vital implications in temozolomide-treated glioblastoma patients [71]. Other in silico studies have reported the favourable prognostic association of CD56^dim^ NK [5] and activated NK cell [72] signatures for TCGA LGG patients. Our analysis further revealed an additional negative association between the abundance of the cytotoxic CD56^dim^ NK cell signature and the expression of HLA-I transcripts in LGG tumours. We also observed that increased expression of HLA-I transcripts was associated with poor LGG patient survival, whereas patients whose tumours exhibited a higher abundance of the CD56^dim^ NK cell signature had a better LGG prognosis.

In contrast to this strong negative HLA-I/CD56^dim^ NK cell relationship, we observed a weak association between HLA-I expression and the infiltration of mature CD8^+^ T cell subsets, which may highlight the importance of CD56^dim^ NK cell-mediated anti-tumour immunity. On the other hand, glioma cells can potentially suppress CD56^dim^ NK cell-mediated immunosurveillance through increased HLA-I abundance while evading CD8^+^ T cells. The relatively limited scope of T cell responses to LGGs compared to higher-grade glioblastomas has been observed in both glioma patient tumours [6,7,8,73] and animal models of gliomas [74,75]. This may be because LGG tumours harbour only a small number of somatic mutations [76], which limits tumour neo-antigen production and subsequent presentation to T cells. However, even gliomas exhibiting hypermutation following temozolomide treatment showed only limited CD8^+^ T-cell infiltration [77,78]. This implies that only limited HLA-I-mediated tumour neoantigen detection by CD8^+^ T cells may take place in LGG, meaning that the overexpression of HLA-I may not provoke a CD8^+^ T response in LGG tumours; this leaves HLA-I overexpression open as a potential immune escape mechanism from NK cell-mediated tumour surveillance. This possibility is supported by research showing that glioblastomas overexpressing HLA-I have a suppressive impact on the tumour-infiltrating NK cell response [79,80].

While assessing the prognostic implications of the NK cell receptors in LGGs, the NKG2x receptors that interact with HLA-E as a ligand displayed strong associations with LGG patient survival, indicating a potential role for the NKG2x:HLA-E signalling axis in the LGG TME. Numerous studies have highlighted the negative clinical implications of aberrant HLA-E expression by various solid cancers, hinting towards the possible immunosuppression of HLA-I-independent recognition and killing by tumour-infiltrating NK cells [81,82,83,84]. Abundant NK-inhibitory signals via HLA-I/NKG2A receptor binding can overwhelm activating signalling through NKG2C/NKG2E [85,86]. Furthermore, the enhanced activation and infiltration of NK cells in the TME following the NKG2A blockade has been described previously [87,88]. Taking this evidence into consideration, it can be speculated that LGG tumours may suppress the activity of beneficial tumour-infiltrating CD56^dim^ NK cells through the HLA-E/CD94-NKG2A signalling axis. However, this immunosuppressive mechanism could be countered using NKG2A/C switch receptor-containing engineered NK cells [89].

## 5. Conclusions

In this study, we identified a consistent and previously unexplored association between the expression of HLA-I molecules and CD56^dim^ NK cells in LGG tumours, highlighting their potential clinical relevance. Our findings suggest that HLA-I molecules may contribute to immune evasion by suppressing tumour-infiltrating cytotoxic NK cells in the glioma microenvironment. While these results offer novel insights into the immunobiology of LGGs, this study is limited by its reliance on transcriptomic data, underscoring the need for experimental validation. Therefore, targeted in vitro and in vivo experiments such as flow cytometry [90], multiplex immunohistochemistry [91,92], and NK-tumour functional co-culture assays [31,93] are critical to gain deeper mechanistic insights into this HLA-I-mediated regulation of NK cells in gliomas.

## Figures and Tables

**Figure 1 cancers-17-01570-f001:**
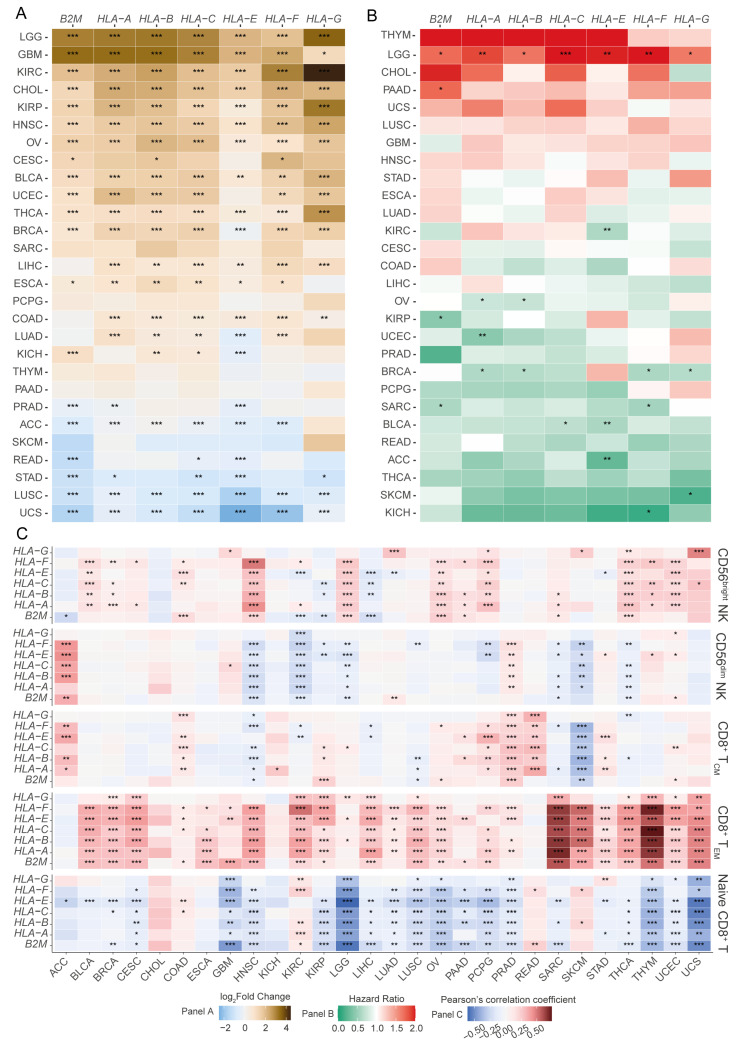
A pan-cancer screen reveals the landscape of altered expression of HLA-I encoding transcripts. (**A**) Differential expression of transcripts encoding HLA-I molecules across solid tumours compared to tumour-adjacent or normal tissues (Wilcoxon signed-ranked test); (**B**) risk stratification of HLA-I molecules using the Cox proportional hazards model; (**C**) association of transcripts encoding HLA-I molecules and the abundance of CD8^+^ T cell subsets (naïve, effector memory, and central memory) and NK cell subsets (CD56^bright^ and CD56^dim^) (Pearson’s correlation coefficient). (*** *p*-value < 0.001, ** *p*-value < 0.01, and * *p*-value < 0.05) (ACC: adrenocortical carcinoma; BLCA: bladder urothelial carcinoma; BRCA: breast invasive carcinoma; CESC: cervical squamous cell carcinoma and endocervical adenocarcinoma; CHOL: cholangiocarcinoma; COAD: colon adenocarcinoma; DLBC: lymphoid neoplasm diffuse large B-cell lymphoma; ESCA: oesophageal carcinoma; GBM: glioblastoma multiforme; HNSC: head and neck squamous cell carcinoma; KICH: kidney chromophobe; KIRC: kidney renal clear cell carcinoma; KIRP: kidney renal papillary cell carcinoma; LAML: acute myeloid leukaemia; LGG: brain lower-grade glioma; LIHC: liver hepatocellular carcinoma; LUAD: lung adenocarcinoma; LUSC: lung squamous cell carcinoma; MESO: mesothelioma; PAAD: pancreatic adenocarcinoma; PCPG: pheochromocytoma and paraganglioma; PRAD: prostate adenocarcinoma; READ: rectum adenocarcinoma; SARC: sarcoma; SKCM: skin cutaneous melanoma; STAD: stomach adenocarcinoma; THCA: thyroid carcinoma; UCEC: uterine corpus endometrial carcinoma).

**Figure 2 cancers-17-01570-f002:**
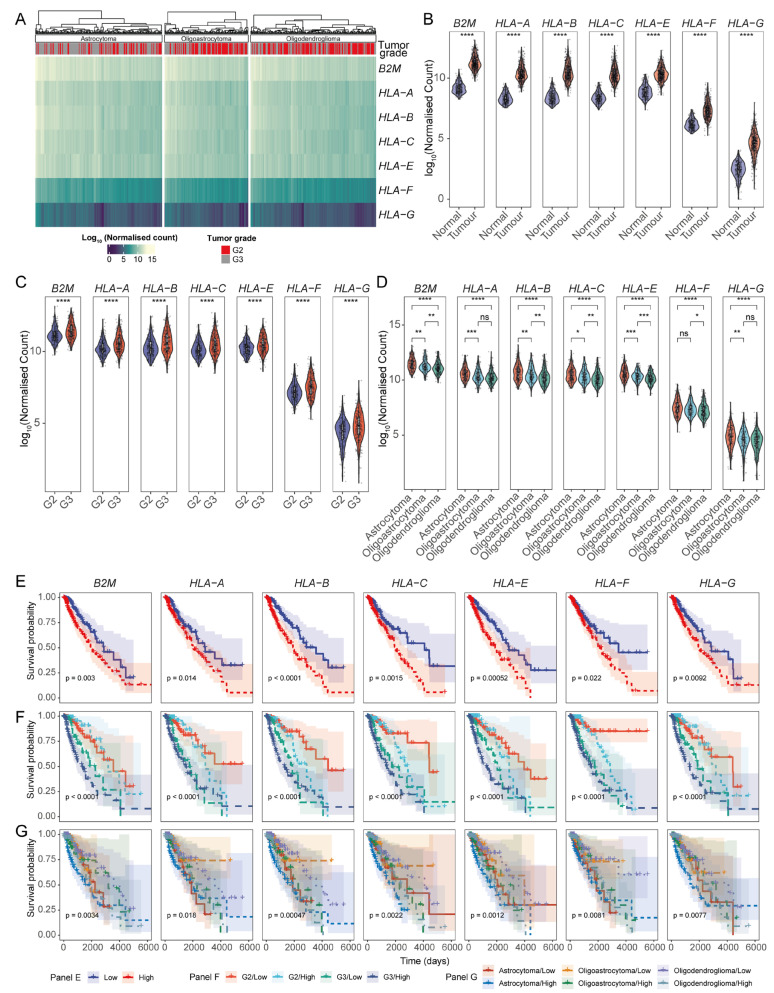
HLA-I-encoding transcripts are aberrantly expressed in TCGA LGG tumours. (**A**). Heatmap of the expression of HLA-I-encoding transcripts in TCGA-LGG patients stratified by glioma grade. (**B**) Violin plots comparing the expression of HLA-I transcripts in tumours versus normal brain tissue; HLA-I transcript abundance across (**C**) different clinical grades and (**D**) glioma molecular subtypes; (**E**) KM survival curves illustrating the prognostic implications of the HLA-I expression in LGGs; Prognostic association of HLA-I expression across (**F**) tumour grades and (**G**) glioma subtypes. (**** *p*-value < 0.0001, *** *p*-value < 0.001, ** *p*-value < 0.01, and * *p*-value < 0.05, ns: non-significant for Wilcoxon signed-ranked test for panels (**B**–**D**)).

**Figure 3 cancers-17-01570-f003:**
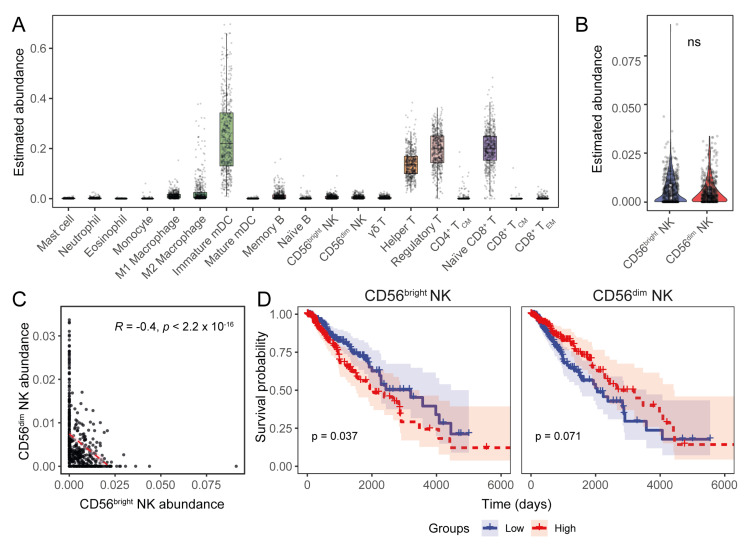
The transcriptional signature of CD56^dim^ NK cells predicts a favourable LGG prognosis. (**A**) Estimated abundance of major immune cell signatures in the LGG TME. (**B**) Comparison of CD56^bright^ and CD56^dim^ NK cell abundances in TCGA LGG transcriptomes. (**C**) Correlation between CD56^bright^ and CD56^dim^ NK cell estimates (Pearson’s correlation coefficient). (**D**) KM curves showing the association of NK subset abundance in LGG patient survival. (ns: non-significant for Wilcoxon signed-ranked test for panel (**B**)).

**Figure 4 cancers-17-01570-f004:**
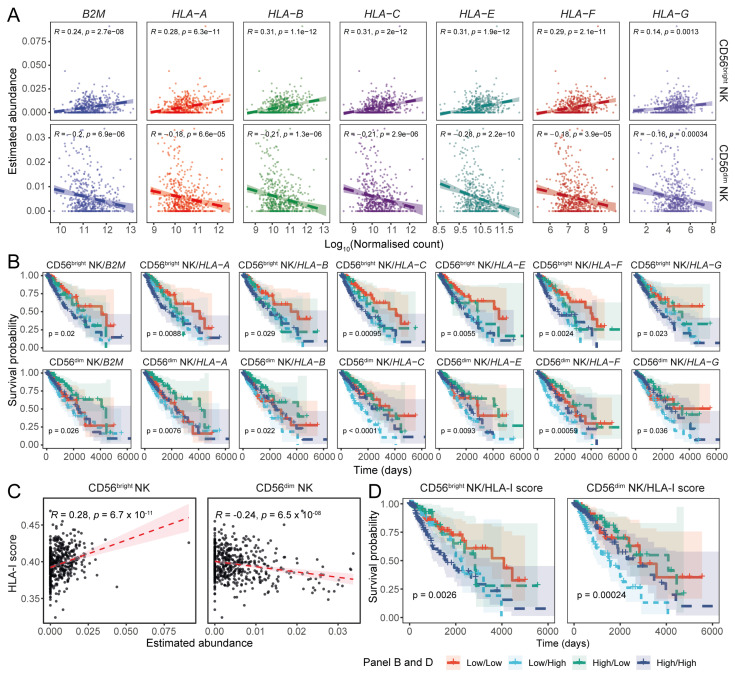
The abundance of HLA-I-encoding transcripts is negatively associated with the transcriptional signature of CD56^dim^ NK cells. (**A**) Scatter plots showing the associations between HLA-I transcript expression and CD56^bright^ and CD56^dim^ NK cell abundances in TCGA LGG tumours (Pearson’s correlation coefficient). (**B**) Combined KM curves illustrating the prognostic implications of HLA-I expression and NK cell abundance in LGGs. (**C**) Correlation between the combined HLA-I scores and estimated NK subset abundances (Pearson’s correlation coefficient). (**D**) Impact of the total HLA-I transcript load on LGG patient survival in the context of NK cell transcriptional signatures.

**Figure 5 cancers-17-01570-f005:**
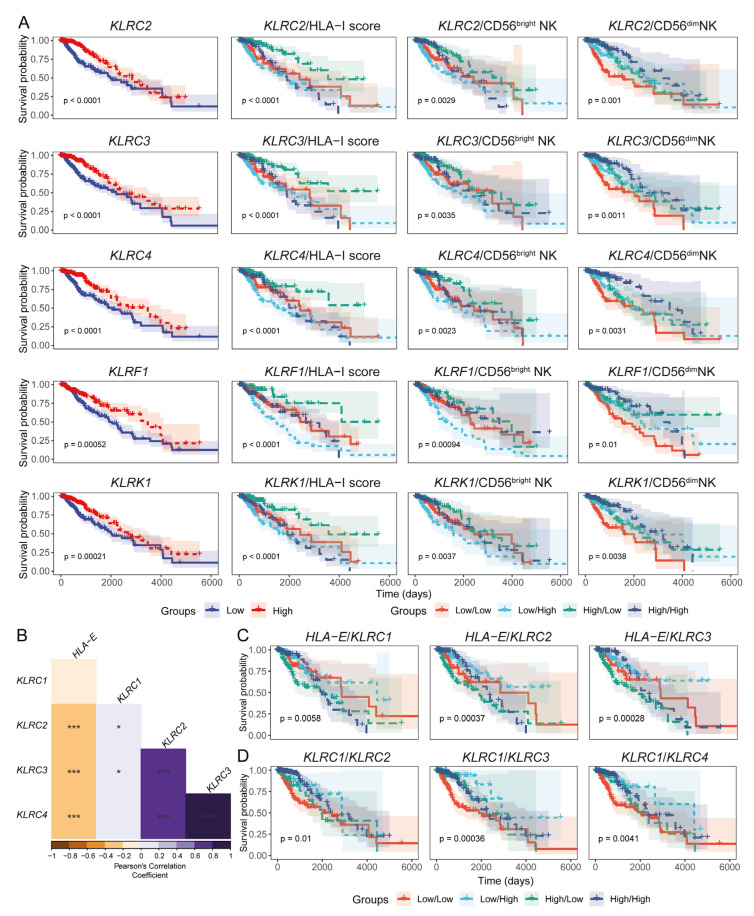
HLA-I expression in LGG tumours is negatively correlated with transcripts encoding activating members of the killer cell lectin-like receptor (KLR). (**A**) Prognostic significance of selected KLRs, both individually and in combination with HLA-I transcript loads and NK subset signatures, in TCGA LGG patients; (**B**). Correlation heatmap illustrating the association between KLRC1 (NKG2A), KLRC2 (NKG2C), KLRC3 (NKG2E), KLRC4 (NKG2F), and HLA-E expression in LGG tumour transcriptomes (Pearson’s correlation coefficient). (**C**) KM curves showing the combined prognostic relevance of HLA-E and transcripts encoding KLR family receptors; (**D**). KM curves demonstrating the combined prognostic impact of activating and inhibitory NKG2x receptor transcripts. (*** *p*-value < 0.001 and * *p*-value < 0.05 for Pearson’s correlation coefficient scores).

**Figure 6 cancers-17-01570-f006:**
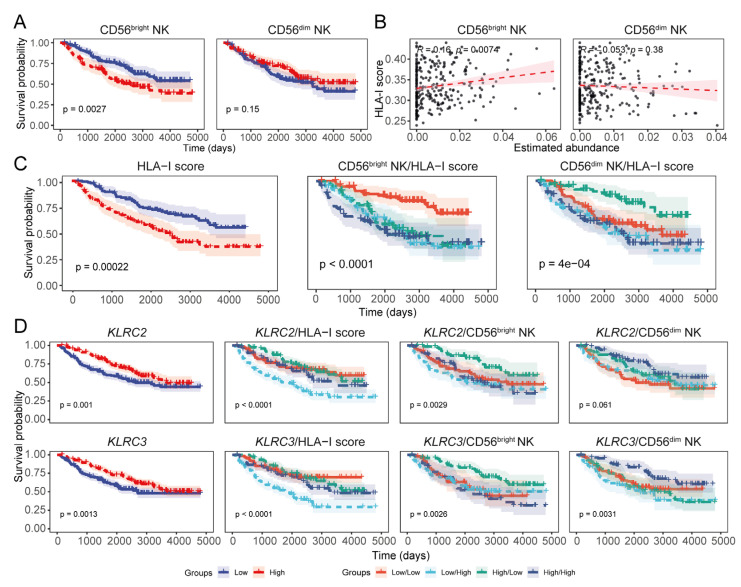
The CGGA LGG data also indicate the negative association between HLA-I transcript expression and CD56^dim^ NK cell abundance. (**A**) KM curves showing the prognostic association of NK subset signatures in CGGA LGG prognoses. (**B**) Correlation between the NK cell subset signatures and the total HLA-I load in CGGA LGG tumours (Pearson’s correlation coefficient). (**C**) Prognostic evaluation of the overall HLA-I transcript load in CGGA LGG patients including its combined effect with the NK subset signatures; (**D**) KM estimates highlighting the prognostic implications of KLRC2 (NKG2C) and KLRC3 (NKG2E) transcript expression in CGGA LGG patients.

## Data Availability

The original contributions presented in the study are included in the article/Appendix A. Further inquiries can be directed to the corresponding authors.

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
