# Peer review of "Reduced HLA-I Transcript Levels and Increased Abundance of a CD56^dim^ NK Cell Signature Are Associated with Improved Survival in Lower-Grade Gliomas"

_cancers, 2025, doi:10.3390/cancers17091570_

Round 1

Reviewer 1 Report

Comments and Suggestions for Authors

Manuscript by Khan et al. provides novel insights into the immune landscape of LGG, particularly the potential inhibitory role of HLA-I on CD56dim NK cells. The study provides valuable insights into the immunogenomic modulation in LGG and presents novel associations with potential clinical relevance. While the study is interesting and well-structured, several aspects require further clarification and improvement. I recommend minor revision before acceptance.

Comments:

  1. The study relies solely on computational analysis of TCGA and CGGA datasets. While the associations are compelling, the authors should acknowledge this limitation more explicitly in the discussion and propose specific in vitro or in vivo experiments to validate their hypotheses (e.g., co-culture of LGG cells with CD56dim NK cells in the context of HLA-I modulation).
  2. The authors use CIBERSORTx and HBCC-derived signatures for CD56bright and CD56dim NK cell deconvolution. However, the validation of these signatures and their specificity in LGG tissue context is unclear. Please clarify how these signatures were tested for robustness in glioma datasets.
  3. The study focuses on NK cells, but CD4+ helper T cells also appear to impact prognosis significantly. Were multivariate survival models including both NK cell subsets and T cell subsets conducted to assess independent contributions?
  4. The scRNA-seq analysis (GSE182109) is mentioned, but the depth is limited. Consider showing representative UMAPs with expression of HLA-I and NK cell receptor genes and validating bulk associations in these single-cell clusters.
  5. Some figure legends lack clarity, especially for composite plots. Add more detail to explain the statistical tests used and sample sizes per group.
  6. Were multiple hypothesis corrections (e.g., Benjamini-Hochberg) applied when assessing prognostic associations across many gene signatures? If not, please justify.
  7. The Simple Summary contains some awkward phrasing. For example, "how the deregulated HLA-I expression associates the clinical outcomes..." should be revised to "how deregulated HLA-I expression is associated with clinical outcomes..."

Author Response

Reviewer #1

Manuscript by Khan et al. provides novel insights into the immune landscape of LGG, particularly the potential inhibitory role of HLA-I on CD56dim NK cells. The study provides valuable insights into the immunogenomic modulation in LGG and presents novel associations with potential clinical relevance. While the study is interesting and well-structured, several aspects require further clarification and improvement. I recommend minor revision before acceptance.

Our response:

We appreciate the feedback from the Reviewer. We have amended the manuscript according to the reviewer’s recommendations.

  1. The study relies solely on computational analysis of TCGA and CGGA datasets. While the associations are compelling, the authors should acknowledge this limitation more explicitly in the discussion and propose specific in vitro or in vivo experiments to validate their hypotheses (e.g., co-culture of LGG cells with CD56dim NK cells in the context of HLA-I modulation).

Our response:

We appreciate the reviewer’s comment on this. We have now acknowledged the limitations of our study in the discussion along with some future directions (lines 407 – 416), which reads as follows:

“In this study, we identified a consistent and previously unexplored association between the expression of HLA-I molecules and CD56dim NK cells in LGG tumours, highlighting their potential clinical relevance. Our findings suggest that HLA-I molecules may contribute to immune evasion by suppressing tumour-infiltrating cytotoxic NK cells in glioma microenvironment. While these results offer novel insights into the immunobiology of LGG, the study is limited by its reliance on transcriptomic data, underscoring the need for experimental validation. Therefore, targeted in vitro and in vivo experiments such as flow cytometry [90], multiplex immunohistochemistry [91, 92] and NK-tumour functional co-culture assays [31, 93] are critical to gain deeper mechanistic insights into this HLA-I-mediated regulation of NK cells in glioma.”

  1. The authors use CIBERSORTx and HBCC-derived signatures for CD56bright and CD56dim NK cell deconvolution. However, the validation of these signatures and their specificity in LGG tissue context is unclear. Please clarify how these signatures were tested for robustness in glioma datasets.

Our response:

We thank the reviewer for this valuable point. The CD56dim NK cell signatures were derived from the HBCC dataset and refined using cellsig (PMID 37952182) to improve robustness. While not experimentally validated in glioma, their application to TCGA and CGGA datasets showed consistent, biologically meaningful trends across tumour grades and subtypes. Similar patterns in the analysed lower grade glioma single-cell data further support their specificity. In Figure S7G–H, we assessed CD56dim NK cell signature abundance in the NK/T cell cluster (cluster 5) relative to other immune cell types. The signature was enriched compared to most, though slightly lower than CD8⁺ T cell subsets, suggesting a mixed NK–T cell population.

  1. The study focuses on NK cells, but CD4+ helper T cells also appear to impact prognosis significantly. Were multivariate survival models including both NK cell subsets and T cell subsets conducted to assess independent contributions?

Our response:

Yes, the multivariate analysis in Figure S5 was performed to assess the independent contributions of the CD56dim NK cell and T cell subsets as well as for the total HLA-I load.

  1. The scRNA-seq analysis (GSE182109) is mentioned, but the depth is limited. Consider showing representative UMAPs with expression of HLA-I and NK cell receptor genes and validating bulk associations in these single-cell clusters.

Our response:

Representative UMAPs showing the expression of HLA-I and NK cell receptor genes are provided in Figure S8.

  1. Some figure legends lack clarity, especially for composite plots. Add more detail to explain the statistical tests used and sample sizes per group.

Our response:

We thank the reviewer for this valuable suggestion. However, we have chosen not to display the number of events on the composite Kaplan–Meier curves, as doing so may reduce clarity and hinder interpretation. To improve transparency, we have updated the Methods section (lines 146–149) to specify the statistical tests used for the Kaplan–Meier analyses. Additionally, we have indicated the specific statistical tests in the relevant figure legends where appropriate.

  1. Were multiple hypothesis corrections (e.g., Benjamini-Hochberg) applied when assessing prognostic associations across many gene signatures? If not, please justify.

Our response:

Yes, multiple hypothesis correction using the Benjamini-Hochberg (BH) method was applied to adjust the global p-values derived from the Kaplan-Meier survival analyses involving multiple variables. This has been clarified in the revised Methods section (lines 146 – 148) and detailed in Supplementary table 1.

  1. The Simple Summary contains some awkward phrasing. For example, "how the deregulated HLA-I expression associates the clinical outcomes..." should be revised to "how deregulated HLA-I expression is associated with clinical outcomes..."

Our response:

Thank you for pointing this out. We have revised the text, which now reads:

“However, how deregulated HLA-I expression impacts clinical outcomes in cancer patients has remained unclear.”

Reviewer 2 Report

Comments and Suggestions for Authors

Manuscript ID: cancers-3525426

Title: Reduced HLA-I transcript levels and increased abundance of a CD56dim NK cell signature correlate with improved survival in low-grade glioma

The present study by Khan et al., The study by Khan et al. relies heavily on bioinformatics, using TCGA, CGGA, and transcriptomic data without experimental validation. While computational analyses provide valuable insights, the lack of wet-lab confirmation makes the findings correlative and speculative, which lessens the reviewer’s interest. Integrating experimental data, particularly on immune cell subsets and HLA-I expression, would strengthen the study’s conclusions. Without this, the manuscript lacks sufficient biological evidence to support its claims. I recommend Rejection it in its current form. Below are key areas that need further investigation to enhance scientific rigor and support the study’s conclusions.

Weakness:

  • Lack of experimental validation
  • Limited mechanistic insights – lack underlying biological mechanisms of association between HLA-I expression and NK cell subsets
  • Inconsistent correlations in different datasets - correlations between HLA-I expression and NK cell subsets differ between the two datasets, which weakens the robustness of the conclusions
  • Potential lack of statistical rigor – For e.g. assessment of confounding factors such as genetic heterogeneity, or other immune subtypes
Comments on the Quality of English Language
  1. There are numerous grammar and syntax errors throughout the manuscript. It is advisable to consult an editing service for a thorough review and clarity. Some examples of errors includes
  • Change “remained elusive” to “remains unclear or has been poorly understood.”
  • Change “various solid cancers” to more specific type, such as “solid tumors, including gliomas”.
  • The statement…“suggesting potential HLA-I modulation of the tumor-infiltrating cytotoxic CD56dim NK cell subset” is not clear. Please rephrase it.
  • The statement…“approximately between 0.25 to 0.75 per 100,000 individuals develops different forms of LGG globally [2, 3]”. Please restructured it for better readability.
  1. Briefly elaborate with reference specifying which NK receptors are being negatively affected by HLA-I
  2. The author stated… “This association varied depending on the histopathological type”. Please mention which specific histopathological types.
  3. The term "normal tissue" and "healthy tissue" are sometimes used interchangeably. Choose one term consistently.
  4. Many sentences in the manuscript are lengthy; please shorten them for clarity.

Author Response

Reviewer #2

The present study by Khan et al., The study by Khan et al. relies heavily on bioinformatics, using TCGA, CGGA, and transcriptomic data without experimental validation. While computational analyses provide valuable insights, the lack of wet-lab confirmation makes the findings correlative and speculative, which lessens the reviewer’s interest. Integrating experimental data, particularly on immune cell subsets and HLA-I expression, would strengthen the study’s conclusions. Without this, the manuscript lacks sufficient biological evidence to support its claims. I recommend Rejection it in its current form. Below are key areas that need further investigation to enhance scientific rigor and support the study’s conclusions.

Our response:

We respectfully disagree with the reviewer’s recommendation to reject. This study is the first to integrate a broad range of transcriptomic analyses to demonstrate the importance of NK cells, NK cell receptors, and MHC class I transcript expression in shaping clinical outcomes in lower grade gliomas. Our conclusions are appropriately scoped and grounded in the data and we make no claims beyond the title, or the analyses presented.

The reliance on bioinformatics is a strength, not a limitation, given the rarity of lower grade gliomas and the size of the datasets utilised (e.g., TCGA: n = 516 patients; CGGA: ~300 patients per cohort). Reproducing a study of this scale with prospectively recruited patients would be prohibitively difficult and ethically challenging. Moreover, there is currently a lack of reliable NK cell-specific tissue markers suitable for immunohistochemistry for any cancer, especially in the brain (e.g., CD56/NCAM-1 is expressed widely in brain), making multiplex validation approaches both technically unfeasible and impractical for a basic science group, such as ours.

Importantly, this work:

  • Provides clinically relevant insights into a poorly understood arm of glioma immunity
  • Identifies novel prognostic associations that could inform future therapeutic strategies
  • Offers a generalisable framework for NK cell transcriptomics in cancer and potentially other diseases.
  • Maximises the value of public datasets in alignment with open science goals.
  • Lays the groundwork for future experimental studies using specific antibodies to tissue NK cell markers, as they become available.

We recognise that experimental validation has value and fully support future studies that would build on these findings along the above lines. However, rejecting a novel, well-executed study simply because it does not include data that would be extraordinarily difficult to generate - especially by a non-clinical group - is, respectfully, unjustified. We believe this work makes a meaningful contribution to the field and should be considered for publication on its merits.

Weakness:

Lack of experimental validation

Our response:

While we acknowledge the absence of experimental validation, our study aimed to identify robust, biologically relevant associations using multiple independent bulk and single-cell transcriptomic datasets. The consistency of our findings across TCGA, CGGA, and single-cell data supports their validity. We have now clearly stated this limitation in the Discussion (lines 407 – 416) and outlined potential future experiments (e.g., flow cytometry, IHC, co-culture assays) to validate these results, which now reads –

“In this study, we identified a consistent and previously unexplored association between the expression of HLA-I molecules and CD56dim NK cells in LGG tumours, highlighting their potential clinical relevance. Our findings suggest that HLA-I molecules may contribute to immune evasion by suppressing tumour-infiltrating cytotoxic NK cells in glioma microenvironment. While these results offer novel insights into the immunobiology of LGG, the study is limited by its reliance on transcriptomic data, underscoring the need for experimental validation. Therefore, targeted in vitro and in vivo experiments such as flow cytometry [90], multiplex immunohistochemistry [91, 92] and NK-tumour functional co-culture assays [31, 93] are critical to gain deeper mechanistic insights into this HLA-I-mediated regulation of NK cells in glioma.”

Limited mechanistic insights – lack underlying biological mechanisms of association between HLA-I expression and NK cell subsets

Our response:

We appreciate the reviewer’s interest in the mechanistic underpinnings of the observed associations. While our study is not designed to directly test mechanistic hypotheses through functional assays, we respectfully disagree with the assertion that it lacks mechanistic insight. Our analyses reveal a consistent and statistically significant negative correlation between the abundance of the CD56dim NK cell signature and MHC class I transcript levels across the TCGA and CGGA cohorts.

This is biologically meaningful, as CD56dim NK cells are the primary cytotoxic subset in humans, and their activity is known to be regulated through inhibitory receptors that recognise MHC-I molecules. The observed inverse relationship provides transcriptomic evidence from real patient datasets supporting the canonical mechanism whereby reduced MHC-I expression facilitates increased NK cell infiltration or activation - a concept well established in the immunology literature but previously unvalidated at scale in gliomas.

Thus, our findings offer an important mechanistic link between low MHC-I expression and increased NK cell involvement in the glioma microenvironment, grounded in human transcriptomic data. We believe this lays a foundation for future studies, including functional investigations or tissue-level analyses, to further delineate the impact of MHC-I-mediated immune evasion and NK cell-driven immunosurveillance in gliomas.

Moreover, whilst our study is computational, it highlights several potential mechanisms mediated by NK cell receptor-ligand interactions in lower grade gliomas that may be important for improved patient survival - such as the association of high HLA-E with increased NKG2A (inhibitory) and reduced CD56dim NK cell abundance. Conversely, higher expression of activating NK receptors (e.g., NKG2C/E, CD57, CD62L) alongside low HLA-I levels was linked to better prognosis. These data suggest a possible HLA-E/NKG2A-driven immunosuppressive axis in lower grade gliomas that could be investigated in future studies. We hope the reviewer will appreciate these insights as a valuable step toward uncovering functional mechanisms for anti-tumour immunity in glioma.

Inconsistent correlations in different datasets - correlations between HLA-I expression and NK cell subsets differ between the two datasets, which weakens the robustness of the conclusions

Our response:

We acknowledge that the TCGA and CGGA datasets differ in sample size, sequencing depth, and patient demographics (see supporting figure A) and likely also in microbiome-host interactions, which can modulate gene expression and influence the strength and consistency of observed correlations. We tested whether the overall patterns of association remain consistent across datasets using a Rank-Rank Hypergeometric Overlap (RRHO) analysis, a method that compares gene expression correlations across datasets. We observed significant overlaps in the direction of gene correlations between datasets - meaning genes linked to high HLA-I or CD56dim NK signatures in one dataset tend to show similar patterns in the other (supporting figure C–D). This supports the robustness of our conclusions, despite expected differences between cohorts.

Supporting figure: Comparison of TCGA and CGGA LGG datasets. A. Total read counts per sample show higher library sizes in TCGA than CGGA; B. PCA shows clear separation between CGGA (blue) and TCGA (red) samples based on gene expression; RRHO heatmaps show overlap in genes correlated with C. HLA-I score and D. CD56dim NK signature between datasets. Consistent positive and negative overlaps are seen in the top-right and bottom-left quadrants.

Potential lack of statistical rigor – For e.g. assessment of confounding factors such as genetic heterogeneity, or other immune subtypes

Our response:

To address potential confounding factors, we performed multivariate analyses including tumour mutation burden (as a proxy for genetic heterogeneity), CD8⁺ T cell subsets (key HLA-I-modulated immune cells), and CD56dim NK cells. As shown in Figure S3, CD56dim NK cell abundance remained an independent prognostic factor, supporting the robustness of our findings. This has been outlined in the manuscript (lines 244–247), which now reads:

“In the multivariate analysis, higher tumour mutation burden (TMB) was observed to be strongly associated with poor survival in LGG patients, while CD56dim NK cell abundance was associated with a trend toward improved outcomes. This trend appeared to be independent of TMB and CD8 T cell subset abundances (Figure S3).”

Figure S3: Multivariate analysis showing the association of CD56dim NK cells, CD8 T cell subsets, and tumour mutation burden with relapse-free survival in LGG patients.

There are numerous grammar and syntax errors throughout the manuscript. It is advisable to consult an editing service for a thorough review and clarity. Some examples of errors includes

Our response:

We have carefully proofread the manuscript and corrected grammatical and syntactic errors throughout.

  1. Change “remained elusive” to “remains unclear or has been poorly understood.”

Our response:

We have revised the sentence accordingly, which now reads:

“However, how deregulated HLA-I expression impacts clinical outcomes in cancer patients has remained unclear.”

  1. Change “various solid cancers” to more specific type, such as “solid tumors, including gliomas”.

Our response:

We have updated the text accordingly.

  1. The statement…“suggesting potential HLA-I modulation of the tumor-infiltrating cytotoxic CD56dim NK cell subset” is not clear. Please rephrase it.

Our response:

We rephrased the statement, which now reads:

“[..] suggesting a potential modulatory role of HLA-I on the tumour-infiltrating cytotoxic CD56dim NK cell subset.”

  1. The statement…“approximately between 0.25 to 0.75 per 100,000 individuals develops different forms of LGG globally [2, 3]”. Please restructured it for better readability.

Our response:

We rephrased the statement, which now reads:

“They account for 15-20% of all primary brain cancers, with an estimated global incidence of 0.25 to 0.75 cases per 100,000 individuals annually [2, 3].”

  1. Briefly elaborate with reference specifying which NK receptors are being negatively affected by HLA-I

Our response:

We have now elaborated in the revised text to specify that HLA-I expression—particularly HLA-E—is negatively associated with activating NK receptors such as NKG2C (KLRC2), NKG2E (KLRC3), CD62L (SELL), and CD57 (B3GAT1). These associations are detailed in Figures 5-6 and Figures S6 and S9. Additionally, we have included appropriate reference to highlight the role of these receptors in anti-tumour immunity (see lines 294 – 295, 299–301).

  1. The author stated… “This association varied depending on the histopathological type”. Please mention which specific histopathological types.

Our response:

The association between HLA-I expression and patient prognosis varies across different histopathological glioma subtypes, as described in Results section 3.2 (lines 207–226) and illustrated in Figure 2 and Supplementary Figure S2.

  1. The term "normal tissue" and "healthy tissue" are sometimes used interchangeably. Choose one term consistently.

Our response:

We have replaced the phrase “normal healthy” with “normal” throughout the manuscript for clarity and conciseness.

  1. Many sentences in the manuscript are lengthy; please shorten them for clarity.

Our response:

We have carefully revised the manuscript to improve readability by shortening several lengthy sentences and improving overall clarity.

Reviewer 3 Report

Comments and Suggestions for Authors

In my opinion, the study is well-conducted and interesting. I have just one consideration: it is difficult to consider as low-grade the WHO grade III gliomas, becouse of their worse clinical outcome and prognosis as compared to grade II ones. I suggest to eliminate the term "low-grade" and define the included tumors only as WHO grades II and III, as stated by the 2021 WHO classification.

In my opinion, the present study is worthy for publication.

Author Response

Reviewer #3

In my opinion, the study is well-conducted and interesting. I have just one consideration: it is difficult to consider as low-grade the WHO grade III gliomas, becouse of their worse clinical outcome and prognosis as compared to grade II ones. I suggest to eliminate the term "low-grade" and define the included tumors only as WHO grades II and III, as stated by the 2021 WHO classification.

In my opinion, the present study is worthy for publication.

Our response:

We thank the reviewer for their thoughtful comment.

We agree that WHO grade III gliomas are clinically and biologically distinct from grade II gliomas, as highlighted in the 2021 WHO classification. Whilst the original TCGA grouped WHO grade II and III gliomas under the label “Low-Grade Glioma (LGG)”, the dataset has since been updated and renamed the ‘Lower Grade Glioma’ study to align with the 2021 WHO classification:

https://www.cancer.gov/ccg/research/genome-sequencing/tcga/studied-cancers/lower-grade-glioma-study.

To maintain consistency with both the updated TCGA nomenclature and the current WHO classification, we have adopted the term ‘lower grade glioma’ throughout the manuscript.

Our analysis begins with the full TCGA-LGG cohort, followed by stratification based on grade and histological subtype where appropriate. Using lower grade glioma (LGG) as an umbrella term at the outset provides clarity and coherence in presenting the overall findings. To address the reviewer’s concern, we have added a clarification to the Methods section (lines 111–113), which now reads-

“The TCGA LGG dataset includes patients diagnosed with lower grade gliomas, encompassing WHO grade II and III astrocytomas and oligodendrogliomas [42], as per classifications prior to the 2021 WHO update [43].”

We hope this clarification resolves the concern and thank the reviewer again for their valuable feedback.

Reviewer 4 Report

Comments and Suggestions for Authors

In the manuscript by Khan and Burrows, computational approaches are used to analyse the transcriptional landscape in different types of cancer with focus on HLA-A,B,C,E,F,G/B2m expression levels as well as CD56dim and CD56bright NK cell subsets and CD8+ T cell subsets. RNA expression data retrieved from TCGA and CGGA databases were correlated with survival probabilities obtaiend from the GDC portal. The authors detected a unique transcriptional signature in low grade gliomas (LGG) where aberrantly high levels of MHC-I expression (in particular HLA-E) were correlated with poor prognosis. Conversely, a high abundance of the cytotoxic CD56dim NK cell subset and low levels of LGG MHC-I overexpression (as compared to adjacent normal brain tissue) was associated with better survival probability. Fully consistent with the hypothesis that the cytotoxic activity of NK cells was a hallmark of good prognosis, high levels of the inhibitory NKG2A receptor and its ligand HLA-E were associated with poor survival while high abundance of the activating NKG2C/E NK cell receptor was associated with a more favorable prognosis. Data obtained with the TCGA and CGGA datasets were consistent and, in essence, also confirmed by a single cell RNA-seq data analysis. As far as I can judge, the application of biostatistical methods and their evaluation was appropriate and correct. The manuscript is very well written and contains a large body of valuable and interesting information also beyond the focus of LGG.

Minor points:

Decode the tumor entities listed in Figure 1 A/B

Add analyses of NCR1 and NCR3 and survival probabilities to Figure S5.

Author Response

Reviewer #4

In the manuscript by Khan and Burrows, computational approaches are used to analyse the transcriptional landscape in different types of cancer with focus on HLA-A,B,C,E,F,G/B2m expression levels as well as CD56dim and CD56bright NK cell subsets and CD8+ T cell subsets. RNA expression data retrieved from TCGA and CGGA databases were correlated with survival probabilities obtaiend from the GDC portal. The authors detected a unique transcriptional signature in low grade gliomas (LGG) where aberrantly high levels of MHC-I expression (in particular HLA-E) were correlated with poor prognosis. Conversely, a high abundance of the cytotoxic CD56dim NK cell subset and low levels of LGG MHC-I overexpression (as compared to adjacent normal brain tissue) was associated with better survival probability. Fully consistent with the hypothesis that the cytotoxic activity of NK cells was a hallmark of good prognosis, high levels of the inhibitory NKG2A receptor and its ligand HLA-E were associated with poor survival while high abundance of the activating NKG2C/E NK cell receptor was associated with a more favorable prognosis. Data obtained with the TCGA and CGGA datasets were consistent and, in essence, also confirmed by a single cell RNA-seq data analysis. As far as I can judge, the application of biostatistical methods and their evaluation was appropriate and correct. The manuscript is very well written and contains a large body of valuable and interesting information also beyond the focus of LGG.

Our response:

We are grateful to the reviewer for their assessment and kind support of our manuscript.

Minor points:

  1. Decode the tumor entities listed in Figure 1 A/B

Our response:

We updated the legend of Figure 1, which now reads –

“Figure 1: A pan-cancer screen reveals the landscape of altered expression of HLA-I encoding transcripts. A. Differential expression of transcripts encoding HLA-I molecules across solid tumours compared to tumour-adjacent or healthy normal tissues; B. Risk stratification of HLA-I molecules using the Cox proportional hazards model; C. Association of transcripts encoding HLA-I molecules and the abundance of CD8+ T cell subsets (naïve, effector memory, and central memory) and NK cell subsets (CD56bright and CD56dim). (***p-value < 0.001, **p-value < 0.01 & *p-value < 0.05) (ACC: Adrenocortical carcinoma; BLCA: Bladder Urothelial Carcinoma; BRCA: Breast invasive carcinoma; CESC: Cervical squamous cell carcinoma and endocervical adenocarcinoma; CHOL: Cholangiocarcinoma; COAD: Colon adenocarcinoma; DLBC: Lymphoid Neoplasm Diffuse Large B-cell Lymphoma; ESCA: Esophageal carcinoma; GBM: Glioblastoma mutiforme; HNSC: Head and Neck squamous cell carcinoma; KICH: Kidney Chromophobe; KIRC: Kidney renal clear cell carcinoma; KIRP: Kidney renal papillary cell carcinoma; LAML: Acute Myeloid Leukemia; LGG: Brain Lower Grade Glioma; LIHC: Liver hepatocellular carcinoma; LUAD: Lung adenocarcinoma; LUSC: Lung squamous cell carcinoma; MESO: Mesothelioma; PAAD: Pancreatic adenocarcinoma; PCPG: Pheochromocytoma and Paraganglioma; PRAD: Prostate adenocarcinoma; READ: Rectum adenocarcinoma; SARC: Sarcoma; SKCM: Skin Cutaneous Melanoma; STAD: Stomach adenocarcinoma; THCA: Thyroid carcinoma; UCEC: Uterine Corpus Endometrial Carcinoma)”

  1. Add analyses of NCR1 and NCR3 and survival probabilities to Figure S5.

Our response:

We update Figure S5 (now Figure S6) following the reviewer’s recommendations -

Figure S6: Clinical relevance of NK cell receptors KLRC1 (NKG2A), B3GAT1 (CD57), NCR1 (NKp46), NCR2 (NKp44), NCR3 (NKp30), and SELL (CD62L) in LGG patients, evaluated in combination with HLA-I load and NK subset abundances.

Reviewer 5 Report

Comments and Suggestions for Authors

Many thanks for giving me this opportunity to review this article. Actually, I think this is a very interesting and important article. While I have several comments and suggestions on this paper to validate its robustness.

  1. Whether the diagnosis on low-grade glioma is based on the newly CNS tumor classification guidelines (version 5) as it is quite difficult to define them directly from TCGA and CGGA data.
  2. Actually, this study lack for some wet lab experiments for strengthening these results.
  3. The authors can not only annotate the single-cell dataset with singleR, they should do it manually as well.
  4. The single-cell dataset is of some interest, I strongly suggest the authors do more work around this dataset.

Author Response

Reviewer #5

Many thanks for giving me this opportunity to review this article. Actually, I think this is a very interesting and important article. While I have several comments and suggestions on this paper to validate its robustness.

Our response:

We thank the reviewer for the insightful comments. We have incorporated the reviewer’s suggestions in the revised manuscript.

  1. Whether the diagnosis on low-grade glioma is based on the newly CNS tumor classification guidelines (version 5) as it is quite difficult to define them directly from TCGA and CGGA data.

Our response:

We agree that WHO grade III gliomas are clinically and biologically distinct from grade II gliomas, as highlighted in the 2021 WHO classification. Whilst the original TCGA grouped WHO grade II and III gliomas under the label “Low-Grade Glioma (LGG)”, the dataset has since been updated and renamed the ‘Lower Grade Glioma’ study to align with the 2021 WHO classification:

https://www.cancer.gov/ccg/research/genome-sequencing/tcga/studied-cancers/lower-grade-glioma-study.

To maintain consistency with both the updated TCGA nomenclature and the current WHO classification, we have adopted the term ‘lower grade glioma’ throughout the manuscript and added a clarifying statement to the Methods section (lines 111–113), which now reads:

“The TCGA LGG dataset includes patients diagnosed with lower grade gliomas, encompassing WHO grade II and III astrocytomas and oligodendrogliomas [42], as per classifications prior to the 2021 WHO update [43].”

  1. Actually, this study lack for some wet lab experiments for strengthening these results.

Our response:

We thank the reviewer for the suggestion. Although our study is computational, the consistency across bulk and single-cell RNA-seq data from independent cohorts supports the robustness of our findings. We have acknowledged the lack of experimental validation in the discussion and outlined potential wet lab approaches for future studies (lines 407 – 416), which reads as follows:

“In this study, we identified a consistent and previously unexplored association between the expression of HLA-I molecules and CD56dim NK cells in LGG tumours, highlighting their potential clinical relevance. Our findings suggest that HLA-I molecules may contribute to immune evasion by suppressing tumour-infiltrating cytotoxic NK cells in glioma microenvironment. While these results offer novel insights into the immunobiology of LGG, the study is limited by its reliance on transcriptomic data, underscoring the need for experimental validation. Therefore, targeted in vitro and in vivo experiments such as flow cytometry [90], multiplex immunohistochemistry [91, 92] and NK-tumour functional co-culture assays [31, 93] are critical to gain deeper mechanistic insights into this HLA-I-mediated regulation of NK cells in glioma.”

  1. The authors can not only annotate the single-cell dataset with singleR, they should do it manually as well.

Our response:

We appreciate the reviewer’s insight on this. We also performed the manual identification of the cluster-specific markers to support the annotation of the clusters. We have now provided the list of the significantly differentially expressed genes of each cluster in Supplementary table 2.

  1. The single-cell dataset is of some interest, I strongly suggest the authors do more work around this dataset.

Our response:

We performed few more extra analyses to highlight the potential presence of the CD56dim NK signature within the immune cell clusters. We evaluated the abundance of CD56dim NK cell signature within the identified NK/T cell cluster (cluster 5) relative to the average abundance of all other immune cell types. This analysis suggests that NK cells may represent a prominent component of this cluster. While CD56dim NK cell signature was more abundant in cluster 5 than most other immune cell types, the CD8⁺ T cell subsets showed slightly higher expression, further indicating that this cluster likely represents a mixed population of NK and T cells. We have now presented this data in Figure S7G-H.

Round 2

Reviewer 2 Report

Comments and Suggestions for Authors

To Authors,

I appreciate the authors’ efforts to revise the manuscript and respond to my comments. The first revision has improved clarity in several areas and has addressed some of the concerns I previously raised.  I appreciate the thoughtful response and acknowledge the challenges inherent in studying lower-grade gliomas, especially regarding the availability of tissue samples and reliable NK cell markers for experimental validation. Their efforts to leverage large-scale public datasets and contribute to open science are commendable.

However, I maintain my original recommendation for rejection in its current form. While bioinformatics analyses can offer valuable insights, they remain correlative and require at least some degree of biological validation to substantiate mechanistic claims—particularly when proposing novel prognostic markers or therapeutic frameworks. The absence of any wet-lab validation limits the study’s translational value and weakens the overall impact of the conclusions.

I encourage the authors to consider complementing their robust computational approach with even minimal in vitro or ex vivo validation where feasible or to more clearly frame the manuscript as exploratory and hypothesis-generating rather than definitive.

Thank you for the opportunity to review this work.

Author Response

To Authors,

I appreciate the authors’ efforts to revise the manuscript and respond to my comments. The first revision has improved clarity in several areas and has addressed some of the concerns I previously raised.  I appreciate the thoughtful response and acknowledge the challenges inherent in studying lower-grade gliomas, especially regarding the availability of tissue samples and reliable NK cell markers for experimental validation. Their efforts to leverage large-scale public datasets and contribute to open science are commendable.

However, I maintain my original recommendation for rejection in its current form. While bioinformatics analyses can offer valuable insights, they remain correlative and require at least some degree of biological validation to substantiate mechanistic claims—particularly when proposing novel prognostic markers or therapeutic frameworks. The absence of any wet-lab validation limits the study’s translational value and weakens the overall impact of the conclusions.

I encourage the authors to consider complementing their robust computational approach with even minimal in vitro or ex vivo validation where feasible or to more clearly frame the manuscript as exploratory and hypothesis-generating rather than definitive.

Thank you for the opportunity to review this work.

Our response:

Thank you for your thoughtful feedback on our revised manuscript. We appreciate your recognition of the improvements made in response to your earlier concerns.

While we understand and respect your suggestion regarding the inclusion of experimental validation, we believe the manuscript presents a valuable exploratory analysis of lower-grade gliomas, with clear implications for future research. Once again, we would like to highlight that the focus of our study is on large datasets, which are most appropriate for predicting patient survival, gene expression, and related outcomes. We have also carefully revised the manuscript to highlight the explanatory nature of the study in the abstract and conclusion sections.

We hope these revisions effectively address your concerns, and we are grateful for your continued engagement with our work.

Reviewer 5 Report

Comments and Suggestions for Authors

Thank you for your invitation to re-review the revised paper. While I think the authors have already addressed all my concerns, so I suggest a publication on this paper.

Thank you.

Best and cheers,

Binghao

Author Response

Thank you for your invitation to re-review the revised paper. While I think the authors have already addressed all my concerns, so I suggest a publication on this paper.

Our response:

Thank you for taking the time to re-review our manuscript. We greatly appreciate your constructive feedback throughout this process, and we’re pleased to hear that the revisions have addressed your concerns.

We are grateful for your recommendation for publication and your continued support of our work.